

**The Effect of the 2013-2016 High Temperature Anomaly in the Subarctic Northeast Pacific**
**(The "Blob") on Net Community Production**
Bo Yang[1], Steven R. Emerson[1], M. Angelica Peña[2]
[1] School of Oceanography, University of Washington, Seattle, WA 98195, USA
[2] Institute of Ocean Sciences, Fisheries and Oceans Canada, PO Box 6000, Sidney, BC, Canada,
V8L 4B2
Corresponding author: Bo Yang (byang9@uw.edu)
Email address: Steven R. Emerson (emerson@uw.edu), M. Angelica Peña (Angelica.Pena@dfo-
mpo.gc.ca)
Key words: The warm blob, net community production, ocean station Papa





**Abstract**
A large anomalously warm water patch (the "Blob") appeared in the NE Pacific Ocean in
the winter of 2013–14 and persisted through 2016 causing strong positive upper ocean
temperature anomalies at Ocean Station Papa (OSP, 50°N, 145°W). The effect of the
temperature anomalies on annual net community production (ANCP) was determined by upper
ocean chemical mass balances of $O_2$ and DIC using data from a profiling float and a surface
mooring. Year-round oxygen mass balance in the upper ocean (0 to 91–111 m) indicates that
ANCP decreased after the first year when warmer water invaded this area and then returned to
the "pre-blob" value (2.4, 0.8, 2.1, and 1.6 mol C $m^{-2}$ $yr^{-1}$ from 2012 to 2016, with a mean value
of $1.7 \pm 0.7$ mol C $m^{-2}$ $yr^{-1}$). ANCP determined from DIC mass balance has a mean value that is
similar within the errors as that from the $O_2$ mass balance but without significant trend (2.0, 2.1,
2.6, and 3.0 mol C $m^{-2}$ $yr^{-1}$ with a mean value of $2.4 \pm 0.6$ mol C $m^{-2}$ $yr^{-1}$). This is likely due to
differences in the air-sea gas exchange, which is a major term for both mass balances. Oxygen
has a residence time with respect to gas exchange of about one month while the $CO_2$ gas
exchange response time is more like a year. Therefore the biologically induced oxygen saturation
anomaly responds fast enough to record annual changes whereas that for $CO_2$ does not.
Phytoplankton pigment analysis from the upper ocean show lower chlorophyll-*a* concentrations
and greater relative abundance of picoplankton in the year after the warm water patch entered the
area than in previous and subsequent years. Our analysis of multiple physical and biological
processes that may have caused the ANCP decrease after warm water entered the area suggests
that it was most likely due to changes in plankton community composition.



## 1 Introduction

Net community production (NCP) in the upper ocean is defined as net organic carbon

production, which equals biological production minus respiration. At steady state when

integrated over a period of at least one year, the annual NCP (ANCP) is equivalent to the flux of

biologically-produced organic matter from the upper ocean to the interior. Both biological

production and respiration processes are temperature dependent, and heterotrophic activities such

as community respiration and zooplankton grazing are usually considered to be more sensitive to

temperature change than autotrophic production (Allen et al., 2005; Brown et al., 2004; Gillooly

et al., 2001; López-Urrutia et al., 2006; Regaudie-De-Gioux and Duarte, 2012; Rose and Caron,

2007). This implies that rising temperature should lead to enhanced heterotrophy and lower NCP

(López-Urrutia et al., 2006).  In contrast, it has also been suggested (e.g., Chen and Laws, 2017)

that the main effect of temperature on community metabolism is likely due to differences in

phytoplankton community composition (e.g. cyanobacteria dominate in warm, oligotrophic

waters, whereas diatoms dominate in cold, nutrient-rich areas) rather than to lower temperature

sensitivity of phytoplankton production.

From  the winter of 2013, a large anomalously warm water patch (the "Blob") appeared

in the NE Pacific Ocean (Bond et al., 2015). The "Blob" had stretched from Alaska to Baja

California by the end of 2015 (Di Lorenzo and Mantua, 2016) and caused widespread changes in

the marine ecosystem, such as geographical shifts of plankton species, harmful algal blooms, and

strandings of fishes, marine mammals, and seabirds (Cavole et al., 2016). Here we calculate the

ANCP with upper ocean oxygen ($O_2$) and dissolved inorganic carbon (DIC) mass balances using

data from Ocean Station Papa in the NE Pacific (OSP, 50°N, 145°W, Figure 1), to determine if

there were significant NCP changes during the anomalous warm event. The monthly Sea Surface



Temperature Anomaly (SSTA) at OSP from 2012 to 2016 (Figure 2) indicates that for most of

the 1ˢᵗ year (starting from June 2012) sea surface temperature (SST) was lower than usual, but

then transitioned to strong positive temperature anomaly from 2013 to 2014. The positive

anomaly continued with a magnitude of ~ 2$^o$C to June 2015, and then dropped back to "normal"

in the summer of 2016.

Our field location is in the subarctic northeast Pacific Ocean at OSP, where repeat

hydrographic cruises have been carried out since 1981 by Fisheries and Oceans Canada with a

frequency of two to three times per year (Freeland, 2007). A NOAA surface mooring has been

deployed at OSP since 2007, for physical and biogeochemical measurements such as

temperature, salinity, wind, ocean current, radiation, oxygen and total gas pressure, pH, and

carbon dioxide ($CO_2$) (Emerson et al. 2011; Cronin et al. 2015; Fassbender et al. 2016). In

addition, Argo profiling floats have been  deployed near OSP since the 2000s (Freeland and

Cummins, 2005). The first floats measured only temperature, salinity, and pressure but then

measurements of oxygen and nitrate were added (Bushinsky and Emerson, 2015; Johnson et al.,

2009). NCP at OSP has been determined using various approaches over the years, including

bottle incubations (Wong, 1995), $^{234}$Th methods (Charette et al., 1999), carbon/nutrient

drawdown (Fassbender et al., 2016; Plant et al., 2016; Takahashi et al., 1993; Wong et al., 2002a,

2002b), and oxygen mass balance (Bushinsky and Emerson, 2015; Emerson, 1987; Emerson et

al., 1991, 1993; Giesbrecht et al., 2012; Juranek et al., 2012; Plant et al., 2016).

**2 Methods**

**2.1 Measurements of O$_2$, DIC, and phytoplankton biomass**

Autonomous in situ oxygen measurements were made on a profiling float deployed by

the University of Washington (Special Oxygen Sensor Argo float, SOS-Argo F8397, WMO #



5903743, Figure 1). The complete dataset is available at
https://sites.google.com/a/uw.edu/sosargo/, and some of the data have been published previously
by Bushinsky and Emerson (2015) and Yang et al. (2017). Oxygen measurements on the SOS-
Argo float were obtained using an Aanderaa optode oxygen sensor with air-calibration
mechanism (Bushinsky et al., 2016) capable of providing the air-sea difference in oxygen
concentration with an accuracy of about $\pm$ 0.2 % and a vertical resolution of 3-5 m in the top 200
m of water column. This float was operated at a cycle interval of ~ 5 days covering depths from
surface to 1800 m.
Partial pressure of seawater $CO_2$ ($pCO_2$), temperature, and salinity data were obtained
from the NOAA mooring at OSP (WMO # 4800400). The complete dataset is available at
http://cdiac.ornl.gov/oceans/Moorings/Papa_145W_50N.html, and some of the data were
published by Fassbender et al. (2016). DIC was calculated using the total alkalinity (TA)-$pCO_2$
pair in CO2sys program Version 1.1 (van Heuven et al. 2011), where TA was calculated using
the linear relationship with salinity developed in Fassbender et al. (2016) (TA $= 37 \times S + 988$)
for the OSP vicinity. The calculation was performed on the total pH scale using the carbonate
dissociation constants ($K_1'$ and $K_2'$) of Lueker et al. (2000), the $HSO_4^-$ dissociation constant from
Dickson et al. (1990), and the $B_T/S$ ratio from Lee et al. (2010). The DIC data were normalized
to the annual mean salinity at OSP (32.5), to eliminate the influence from evaporation/dilution.
Water samples for phytoplankton abundance and community composition were collected
at OSP during 14 Line P repeat hydrographic cruises aboard the CCGS John P. Tully from 2012
to 2016 (February, June, and August for each year). Phytoplankton biomass, measured as total
chlorophyll $a$ (chl-$a$) concentrations, and the contribution of the main taxonomic groups of
phytoplankton to chl-$a$ were determined from high performance liquid chromatography (HPLC)





measurements of phytoplankton pigment concentrations (chlorophylls and carotenoids, Zapata
et al. 2000) followed by CHEMTAX v1.95 analysis (Mackey et al., 1996). Eight algal groups
were included in the chemotaxonomic analysis: diatoms, haptophytes, chlorophytes,
pelagophytes, prasinophytes, dinoflagellates, cryptophytes, and cyanobacteria. However,
cryptophytes were not found since their biomarker pigment, alloxanthin was not detected in any
of our samples. Pigment ratios for each algal group were obtained from Higgins et al. (2011) and
used as 'seed' values for multiple trials (60 runs) from randomized starting points, as described
by Wright et al. (2009). The same initial pigment ratios (Table 1a) were used in all cruises but
each cruise was run separately to allow potential variations in the CHEMTAX optimization to be
expressed. The range of final pigment ratios are given in Table 1b. The six best solutions (those
with the lowest residuals) were averaged for estimating the taxonomic abundances.

**2.2 Models used for NCP calculation**

**2.2.1 Oxygen mass balance model**

Oxygen, temperature, and salinity data from SOS-Argo F8397 and wind speed ($U_{10}$) data from
NOAA PMEL OSP mooring (https://www.pmel.noaa.gov/ocs/data/disdel/,
https://www.pmel.noaa.gov/ocs/data/fluxdisdel/ ) were used in a multi-layer upper ocean $O_2$
mass balance model to calculate NCP. This model frame (Figure 3) is similar to what was used
in Bushinsky and Emerson (2015), which compartmentalizes the upper ocean (0-150 m) into a
mixed layer box (with variable height) with one meter boxes below. This model assumes that
horizontal processes are not important. Because horizontal gradients of oxygen supersaturation
are small, lateral transport has much less influence on this property than fluxes from air-sea gas
exchange, vertical advection, and diapycnal eddy diffusion. A detailed assessment of this
assumption is given in Yang et al. (2017).





We define ANCP as the flux of organic carbon that escapes the "upper ocean" after a
complete seasonal cycle. To be consistent with this definition NCP is integrated vertically from
the surface ocean to the winter mixed layer depth, which in this location is roughly equal to the
pycnocline depth. Because internal waves cause a 10 to 20 meter variation in the depth of density
surfaces in this location, we used the annual mean pycnocline depth as the base of the modeled
"upper ocean" to conserve mass in the model.  Fluxes across the base of the upper ocean are
calculated using measured gradients in oxygen at the density of the pycnocline, independent of
its depth.
Oxygen concentration changes over time in the modeled "upper ocean" with depth of h
(dh[$O_2$]/dt) are the sum of:  gas exchange fluxes ($F_{A\text{-}W}$), vertical advection flux ($F_V$), diapycnal
eddy diffusion ($F_{Kz}$), entrainment between the mixed layer and the water below ($F_E$), and net
biological oxygen production ($J_{NCP}$).

$$\frac{dh[O_2]}{dt} = F_{A-W} + F_V + F_{Kz} + F_E + J_{NCP} \qquad \text{mol m}^{-2}\,\text{d}^{-1} \quad (1)$$


$F_{A\text{-}W}$ is calculated only for the mixed layer box, using the a gas exchange model that includes
both diffusion and bubble processes (Emerson and Bushinsky, 2016; Liang et al., 2013). With
the time step (3 h) used in our case, the mixed layer change between time steps is always smaller
or equal to 1 m, so entrainment occurs only between the mixed layer box and the box below. The
entrainment flux ($F_E$) that gets out of the mixed layer box ends up going into the box below and
vice versa, so $F_E$ for these two boxes have the same value but different signs and cancel each
other out. $F_V$ is calculated from Ekman pumping rate (derived from wind speed) and oxygen
gradient from SOS-Argo measurements. $F_{Kz}$ is calculated with oxygen gradient and diapycnal
eddy diffusion coefficient from Cronin et al. (2015), which decreases with depth from the base of



the mixed layer to a background value of $10^{-5}$ m$^{-2}$ s$^{-1}$ (Whalen et al., 2012) with a 1/e scaling
described in Sun et al. (2013) (See also Bushinsky and Emerson, 2015). For the mixed layer
reservoir $F_{kz}$ and $F_V$ are considered only at the base of the box. For all the boxes below the mixed
layer, $F_{kz}$ and $F_V$ are considered both on the top and at the base of each box. Biological oxygen
production, $J_{NCP}$, is the difference between the calculated fluxes and the measured time rate of
change (left hand side of Equation 1). This value is converted from oxygen to carbon production
(i.e. ANCP) using a constant oxygen to carbon ratio of 1.45 (Hedges et al., 2002).

The uncertainty of ANCP was estimated using a Monte Carlo approach. Confidence

intervals for oxygen measurements and the gas exchange mass transfer coefficients used in the
oxygen mass balance model were assigned to the model, and varied randomly while ANCP was
calculated in two hundred runs for each calculation. Details of this approach are presented in the
supporting information and Yang et al. (2017).
**2.2.2 DIC mass balance model**

We used a similar mass balance model for DIC, in which the base of the modeled "upper

ocean" is set to the annual mean pycnocline depth (the same as the oxygen mass balance model).
This choice of the upper ocean depth distinguishes this model from the mixed layer model used
in Fassbender et al. (2016).  Fluxes at the base of the upper ocean in our model use DIC
gradients, diapycnal eddy diffusion coefficients, and upwelling velocities determined at the mean
pycnocline depth while Fassbender et al. (2016) used the values at the bottom of the mixed layer.
Because the OSP surface mooring provided only the mixed layer DIC data, we assumed that
there is no annual net DIC change in the depth region between the mixed layer and the annual
mean pycnocline depth.  The depth gradient of DIC used to calculate fluxes across the
pycnocline was calculated from measured oxygen gradients assuming $dO_2/dz$ to $dDIC/dz$ ratio of

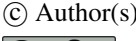



1.45 (Hedges et al., 2002). Thus, we assume for this calculation that the DIC change at the
pycnocline depth is only due to degradation of organic matter, which ignores the change due to
CaCO$_3$ dissolution (Fassbender et al., 2016). For the DIC mass balance the multi-layer model is
equivalent to a one-layer model:

$$\frac{dh[DIC]}{dt} = F_{A-W} + F_V + F_{Kz} + F_E + J_{NCP} \qquad \text{mol m}^{-2}\text{ d}^{-1} \qquad (2)$$


where the DIC change (dh[DIC]/dt) for the modeled upper ocean (the air –sea interface to the
mean depth of the pycnocline) is due to air-water CO$_2$ exchange (F$_{A-W}$) at the air-sea interface,
vertical advection (F$_V$) and diapycnal eddy diffusion (F$_{Kz}$) at the base of the modeled "upper
ocean", and net biological carbon production (J$_{NCP}$) in between. For this one-layer model,
entrainment occurred within the same layer (box) and therefore there is no net entrainment flux
(F$_E$ = 0).  The air-sea gas-exchange mass transfer coefficient is calculated as a function of wind
speed using equations from Wanninkhof (2014). The DIC gradients used for F$_V$ and F$_{Kz}$ are
derived from oxygen gradients at the pycnocline depth as described above.
**2.3 Temperature dependence of NCP derived from the metabolic theory of ecology**

The correlation between NCP variation and environmental temperature could be

attributed to the temperature dependence of planktonic metabolism. Regaudie-De-Gioux and
Duarte (2012) derived the temperature dependences of gross primary production (GPP) and
community respiration (CR) using the metabolic theory of ecology and a large historical dataset
on volumetric planktonic metabolism in different seasons and ocean regimes (1156 estimates of
volumetric metabolic rates and the corresponding water temperature). Equations 3 & 4below are
their linear regressions between the natural logarithm of the specific metabolic rates (*GPP/Chla*
and *CR/Chla*) and the inverted water temperature (*1/kT*),



$$Ln\frac{GPP}{Chla} = a_p\frac{1}{kT} + b_p \tag{3}$$

$$Ln\frac{CR}{Chla} = a_r\frac{1}{kT} + b_r \tag{4}$$


where *Chla* is the  chlorophyll-*a* concentration, $k$ is the Boltzmann's constant, $T$ is the
environmental temperature in Kelvin, and $a_p$, $b_p$, $a_r$, $b_r$ are slopes and intercepts for each linear
regression. The temperature dependence of *GPP/CR* can be derived by combining Equations
3 & 4:

$$\frac{GPP}{CR} = EXP\left[(a_p - a_r)\frac{1}{kT} + (b_p - b_r)\right] \tag{5}$$


Since the community respiration (CR) includes the respiration of both autotrophs and
heterotrophs, NCP can be calculated as the difference between GPP and CR.

$$NCP = GPP - CR = GPP\left(1 - \frac{1}{\frac{GPP}{CR}}\right) \tag{6}$$

Combining Equations 5 and 6 gives us the NCP-temperature relationship.

$$NCP = GPP\left\{1 - \frac{1}{EXP[(a_p - a_r)\frac{1}{kT} + (b_p - b_r)]}\right\} \tag{7}$$


**3 Results**
**3.1 Oxygen and DIC measurements**

The evolutions of density, oxygen concentration, and the oxygen anomaly in percent

supersaturation ($\Delta O_2 = ([O_2]/[O_2]_{sat} - 1) \times 100$) determined by the profiling float at OSP from 2012
to 2016 are presented in Figure 4(a-c). The saturation concentration of oxygen ($[O_2]_{sat}$) was
calculated using equations from Garcia and Gordon (1992, 1993). The thin black line indicates



the mixed layer depth, which is defined by a density offset from the value at 10 m using a
threshold of 0.03 kg m$^{-3}$ (de Boyer Montégut, 2004). The thick blue line indicates the pycnocline
with a density of $\sigma_\theta$ = 25.8 kg m$^{-3}$, which follows [O$_2$] gradients well (Figure 4b). The white
boxes indicate the modeled "upper ocean" for each year, in which base of the modeled "upper
ocean" is the mean pycnocline depth for each year. Oxygen in the mixed layer was
supersaturated from mid April to October/November, and near saturation or slightly
undersaturated for the rest of the year (Figure 4c).

The evolution of salinity normalized DIC in the mixed layer determined by the OSP

mooring is presented in Figure 4d. The $p$CO$_2$ sensor stopped working during two periods in 2013
and 2016 (indicated with dash line boxes), and therefore the data for these two periods is filled
with interpolated values. Strong summertime DIC drawdown was observed in each year with the
lowest DIC around September.

**3.2 Annual Net Community Production**

All the terms of the oxygen mass balance calculation in each year are presented in Table

2a. The ANCP results (2.4 ± 0.6, 0.8 ± 0.4, 2.1 ± 0.4 and 1.6 ± 0.4 mol C m$^{-2}$ yr$^{-1}$, with a mean
value of 1.7 ± 0.7 mol C m$^{-2}$ yr$^{-1}$) indicate that ANCP initially decreased after warmer water
invaded this area (2013-14) and then returned to the "pre-blob" value of 2012-13 in subsequent
years. Given the uncertainty in the estimate  of ANCP in each year, the value during year 2013-
14  is significantly different at the 95% confidence interval (as determined by t-test, Bethea et al.,
1975). With the exception of the unusually low value for 2013-14, ANCP values from oxygen
mass balance calculation are very close to the historical ANCP estimates at OSP (2.3 ± 0.6 mol
C m$^{-2}$ yr$^{-1}$, Emerson 2014).



If we integrate the ANCP from the ocean surface to the depth of the mixed layer
(ANCP$_{\text{mixed layer}}$ in Table 2a) instead of to the annual mean depth of the pycnocline, the results are
higher ( 3.4, 1.3, 2.3 and 2.3 mol C m$^{-2}$ yr$^{-1}$, with a mean value of 2.4 ± 0.9 mol C m$^{-2}$ yr$^{-1}$).
While the mean value is higher because it includes some organic carbon flux that is degraded
between the mixed layer and pycnocline in summer, the annual trend, in which ANCP is
significantly lower in year two (2013-14), is the same as that in which ANCP values were
determined for the depth interval above the pycnocline.
In comparison, ANCP values determined from DIC mass balance are 2.0, 2.1, 2.6, 3.0
mol C m$^{-2}$ yr$^{-1}$, with a mean value of 2.4 ± 0.5 mol C m$^{-2}$ yr$^{-1}$ (Table 2b). The mean value is
similar within the errors of the value determined from the oxygen mass balance (1.7 ± 0.7 mol C
m$^{-2}$ yr$^{-1}$) but there is no significant change between the second year (2013-14) and those before
and after. The somewhat higher value could be due to the assumption we made about DIC
change below the mixed layer or because we neglected horizontal advection (See Discussion).

**3.3 Phytoplankton abundance and community composition**

Chl-*a* concentration, an indicator of phytoplankton biomass, was about 50% lower (0.22
mg m$^{-3}$) during the period from August 2013 to June 2014 than during the rest of the 2012 to
2016  period (Figure 5a) and the historical annual average at OSP(Peña and Varela, 2007). Chl-*a*
resumed to the 2012-13 level in August 2014 and had a significant increase in the summer of
2016. 19'-hexanoyloxyfucoxanthin (Hex), which is mainly derived from prymnesiophytes, was
found to be the most abundant pigment after T-chla (Figure 5b). Fucoxanthin (Fuco), a pigment
associated with diatoms, haptophytes and pelagophytes, was also abundant and showed increased
concentration (0.54 mg m$^{-3}$) in June 2016, coinciding with increased T-chla. After Hex, and
Fuco, chlorophyll-b was the most abundant pigment (0.36 to 0.27 mg m$^{-3}$), indicating the





presence of green algae. We also detected occasionally lutein (0-0.125 mg m$^{-3}$), violaxanthin (0-
0.012 mg m$^{-3}$) and, prasinoxanthin (0-0.005 mg m$^{-3}$), which are biomarkers for green algae.
The CHEMTAX analysis detected the presence of seven classes of phytoplankton
(Figure 5c) and showed an increase in the relative contribution of cyanobacteria and
chlorophytes during the "Blob" period with the highest proportion of the former group in June of
2014 and the latter in June 2015 (Figure 5c). There was also a decrease in the abundance of
diatoms from August 2013 to June 2015. The remainder of the phytoplankton community was
primarily composed of haptophytes and the contribution of the other phytoplankton groups was
variable and showed no consistent year-to-year variability. By August 2015 the phytoplankton
community had returned to a similar relative composition as observed in 2012-13, with
nanoplankton (mostly haptophytes) being dominant and with microplankton (diatoms and
dinoflagellates) increasing in abundance. The input matrix (Table 1a) appeared to describe the
environment well since the final pigment ratio matrix did not differ dramatically from the initial
input values.
**4 Discussion**
**4.1 Comparisons of ANCP from oxygen and DIC mass balances**
Although the ANCP are integrated to the same depth in our oxygen and DIC mass
balance models, as mentioned in Section 3.2, the ANCP determined from DIC mass balance (4-
year mean: 2.4 ± 0.5 mol C m$^{-2}$ yr$^{-1}$) is somewhat higher than the value determined from oxygen
mass balance (4-year mean: 1.7 ± 0.7 mol C m$^{-2}$ yr$^{-1}$), but still within the error of the model.
There are two possible reasons for such discrepancy. First of all, due to the lack of DIC data
below the mixed layer, for the DIC model we made an assumption that there is no annual net
DIC change in the depth region between the mixed layer and the annual mean pycnocline depth.





With this assumption, the ANCP from DIC mass balance is higher because it includes the
organic carbon that is degraded between the mixed layer and pycnocline in summer, so the
ANCP from DIC mass balance (4-year mean: $2.4 \pm 0.5$ mol C m$^{-2}$ yr$^{-1}$) is very similar to the
mixed layer ANCP determined from our oxygen mass balance model (4-year mean: $2.4 \pm 0.9$
mol C m$^{-2}$ yr$^{-1}$) and the mixed layer ANCP determined by Fassbender et al. (2016) ($2 \pm 1$ mol C
m$^{-2}$ yr$^{-1}$). The second possible reason that the 4-year mean value of ANCP determined from the
DIC mass balance is higher than the value determined from the oxygen mass balance is
horizontal advection. Because gas exchange resets the oxygen saturation anomaly for oxygen
about ten times faster than $CO_2$, the DIC mass balance is more vulnerable to horizontal fluxes
than the $O_2$ mass balance. If we assumed that the difference in ANCP estimated from these two
tracers (0.7 mol C m$^{-2}$ yr$^{-1}$) is due to horizontal advection, and calculate the horizontal DIC
gradient using the 4-year mean horizontal velocity at OSP of 0.08 m s$^{-1}$, we found that a
horizontal DIC gradient of $1 \times 10^{-8}$ mol m$^{-4}$ is required to cause the difference of 0.7 mol C m$^{-2}$ yr$^{-}$
$^{1}$, which is possible at this location (horizontal DIC gradient along the 4-year mean horizontal
flow at OSP is about $2 \sim 3 \times 10^{-8}$ mol m$^{-4}$ from GLODAP v1.1 gridded product, Key et al., 2004).

As for the inter-annual changes in ANCP, the oxygen mass balance calculation shows

that ANCP had a significant decrease in 2013-14 and then returned to the "pre-blob" level in the
following years whereas ANCP calculated from DIC mass balance does not show this trend.
Since air-sea exchange is a large part of the flux mass balance for both oxygen and $CO_2$ (Table
2), a likely reason for this discrepancy is due to the shorter residence time with respect to gas
exchange for the oxygen compared to the $CO_2$ saturation anomalies. An example of the residence
time calculation is included in the supporting information where it indicates that the gas
exchange residence time in the upper ocean for oxygen is about one month and that for $CO_2$ is



about one year (See also Emerson and Hedges, 2008, Chapter 11). Thus, the biologically induced
saturation anomaly for oxygen responds fast enough to record annual changes whereas that for
$pCO_2$ and DIC does not. One the other hand, as discussed above, since DIC mass balance is more
vulnerable to horizontal flux than oxygen mass balance, the DIC signal might already been
"smoothed" by the horizontal flux, which may also explain why the inter-annual ANCP changes
were not observed by using the DIC mass balance approach. The sharp decrease in ANCP from
the oxygen mass balance in 2013-14 is consistent with the decrease in chlorophyll concentration
by about 50% observed for the same period (Figure 5a). Hence, from this point forward we will
focus on analyzing the factors that might influence ANCP variations determined by the oxygen
mass balance model.
**4.2 Causes of ANCP decrease**
In the following paragraphs, we analyze connections between ANCP decrease and the
"Blob" temperature anomaly in the context of multiple physical and biological processes,
including the choices of start time from which ANCP are calculated, the base depth of the
modeled "upper ocean", planktonic metabolism, and changes in phytoplankton community
composition.
Our observations began in June 2012, 10 – 12 months before the positive SST anomalies.
To determine whether the start date for determining the ANCP values affects the results, we
began the time series on four different months (Table 3).  We are somewhat limited because
there is only about 12 "pre-blob" months before June, 2012.  However, as shown in Table 3, as
long as there are more "pre-blob" months than "Blob-affected" months in the 1[st] year, the
significant ANCP decrease from 1[st] to 2[nd] year is still observed and the trend of ANCP variation
for those 4 years remains.




To determine whether the annual mean pycnocline depth (the white rectangles in Figure
4a-4c) influences the ANCP trends we calculated ANCP using the 4-year mean depth of 100 m
for the modeled "upper ocean". The ANCP results only change slightly (2.6, 1.0, 1.9, and 1.6
mol C m$^{-2}$ yr$^{-1}$) and the decrease in 2013-14 is still statistically significant, indicating  that the
different base depth used for the modeled "upper ocean" is not the key factor that causes ANCP
changes.
To test if the temperature dependence of planktonic metabolism is strong enough to
cause the ANCP decline we observed (e.g. 1.6 mol C m$^{-2}$ yr$^{-1}$ between 2012-13 and 2013-14), we
calculated the GPP from measured NCP of year 1 (2012-13) using Equation 7, and assumed GPP
was constant for all four years so we could then determine the effect of temperature on NCP
based on the metabolic theory of ecology (Equation 7).  Since the specific phytoplankton growth
rate increases with increasing temperature (e.g. Regaudie-De-Gioux and Duarte, 2012; Chen and
Laws, 2017), if phytoplankton biomass would have remained the same during the "blob", GPP
would have increased. Thus, assuming a constant GPP in this calculation is somewhat
speculative, but it at least provides a first order assessment of the metabolic temperature effect on
ANCP. The parameterizations derived with datasets from Arctic were used (Regaudie-De-Gioux
and Duarte, 2012), because it gives the largest change in ANCP. The results (Table 4) indicate
that temperature dependence of planktonic metabolism is not strong enough to account for the
measured ANCP decrease in the 2$^{nd}$ year (2013-14), suggesting that this is not the major reason
for the observed ANCP decline.
Having ruled out the above likely candidates, we suggest that the low phytoplankton
biomass observed in the 2$^{nd}$ year (2013-14, Figure 5a), and the observed change in phytoplankton
community composition (Figure 5c) are the most likely causes for the ANCP decrease. In



general, larger phytoplankton (i.e. microplankton) are more efficient exporters than smaller
nanoplankton and picoplankton (e.g., Chen and Laws, 2016). Given the lower export rates of
picoplankton (e.g. cyanobacteria) than those of larger phytoplankton (e.g. diatoms) the observed
changes in phytoplankton community composition (Figure 5b) in 2013-14, which included a
decrease in the relative abundance of diatoms, and an increase in the relative abundance of
cyanobacteria and green algae (chlorophytes), could have further contributed to the decrease in
ANCP. After the initial response to the temperature anomaly, chl-$a$ concentration and the
phytoplankton community composition returned to a level similar to those observed before the
warming occurred, suggesting that the plankton community rapidly adapted to the higher
temperature.
**5 Conclusions**
The annual net community production (ANCP) at Ocean Station Papa (OSP) in the
subarctic Northeast Pacific Ocean was determined from June 2012 to June 2016 to examine the
effect of the temperature anomaly on the efficiency of carbon export. The ANCP determined by
oxygen mass balance had a four year mean value of $1.7 \pm 0.7$ mol C m$^{-2}$ yr$^{-1}$, whereas ANCP
determined by DIC mass balance gives a somewhat higher mean value ($2.4 \pm 0.5$ mol C m$^{-2}$ yr$^{-1}$).
ANCP for individual years determined from $O_2$ mass balance showed a significant decrease in
year 2 (2013-14) after the onset of the temperature anomaly, but no significant decrease in
ANCP was found when calculated with DIC mass balance. We believe that this indicates that
the DIC concentration and $p$CO$_2$ respond too slowly to capture annual changes in ANCP. Based
on our observations and historical ANCP estimates at OSP as reference, we found there was a
significant ANCP decrease in 2013-14 due to the warm anomaly, which is consistent with the
findings from concurrent phytoplankton data. Possible mechanisms for the observed decrease in



ANCP by the oxygen mass balance in the second year were analyzed in the context of multiple
physical and biological processes that could be affected by temperature anomaly. Our analysis
showed that the ANCP decrease was most likely due to changes in phytoplankton abundance and
community composition after the "Blob" entered the area.

*Data availability.*
Float data are available online (https://sites.google.com/a/uw.edu/sosargo/home). Mooring data
is available online at: http://cdiac.ornl.gov/oceans/Moorings/Papa_145W_50N.html.
*Author contributions.*
BY and SRE designed the experiments. BY developed the model code and process the data. AP
provided the phytoplankton data. BY and SRE prepared the manuscript with contributions from
all co-authors.
*Competing interests.*
The authors declare that they have no conflict of interest.
*Acknowledgements.*
We thank Dr. Stephen Riser and Dana Swift for their assistance in development of the SOS-Argo
float, and scientists of NOAA PMEL and the Institute of Ocean Sciences (IOS) and crews of
CCGS John P. Tully, for their work on OSP mooring and Line P cruises. Special thanks are
given to Dr. John Crusius for the constructive discussions and comments on this study. This
work was supported by National Science Foundation grant OCE-1458888.

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



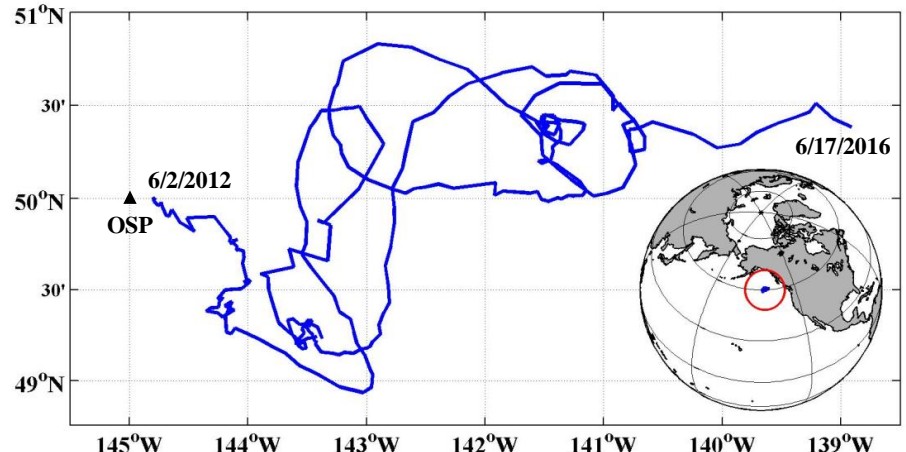

**Figure 1** Study area and float path from 2012 to 2016. The black triangle indicates the position
of Ocean Station Papa (OSP) Mooring, and the blue line indicates the trajectory of the SOS-Argo
float which was within roughly a $2^o$ (N-S) $\times 6^o$ (E-W) box.

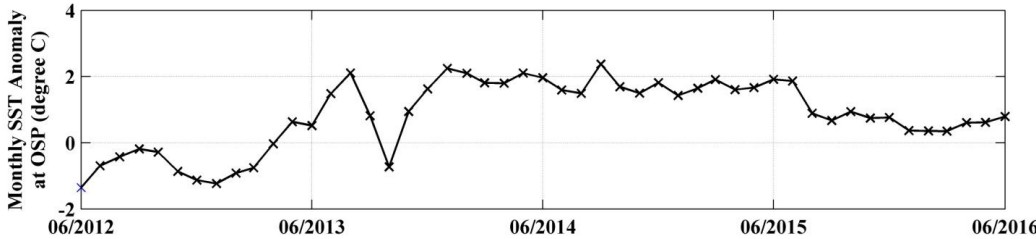

**Figure 2** Monthly SST Anomaly at Ocean Station Papa (OSP). The anomaly is defined as the
difference between the measured SST and the mean of 1971-2000. Data are from:
http://iridl.ldeo.columbia.edu/maproom/Global/Ocean_Temp/Anomaly.html



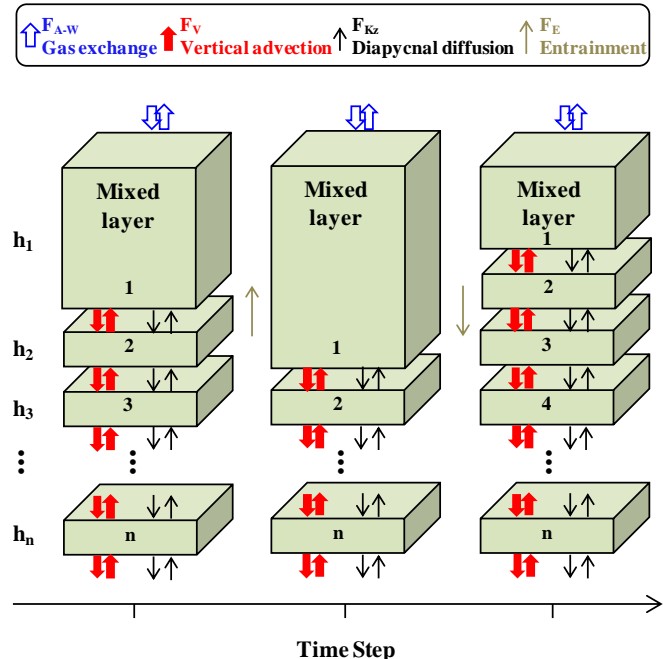


**Figure 3** Schematic of the multi-layer upper ocean oxygen mass balance model (adapted from
*Bushinsky and Emerson*, 2015). Fluxes (F) are from air-sea gas exchange ($F_{A-W}$, including
diffusion and bubble processes), vertical advection ($F_V$), diapycnal eddy diffusion ($F_{Kz}$), and
entrainment ($F_E$).

558





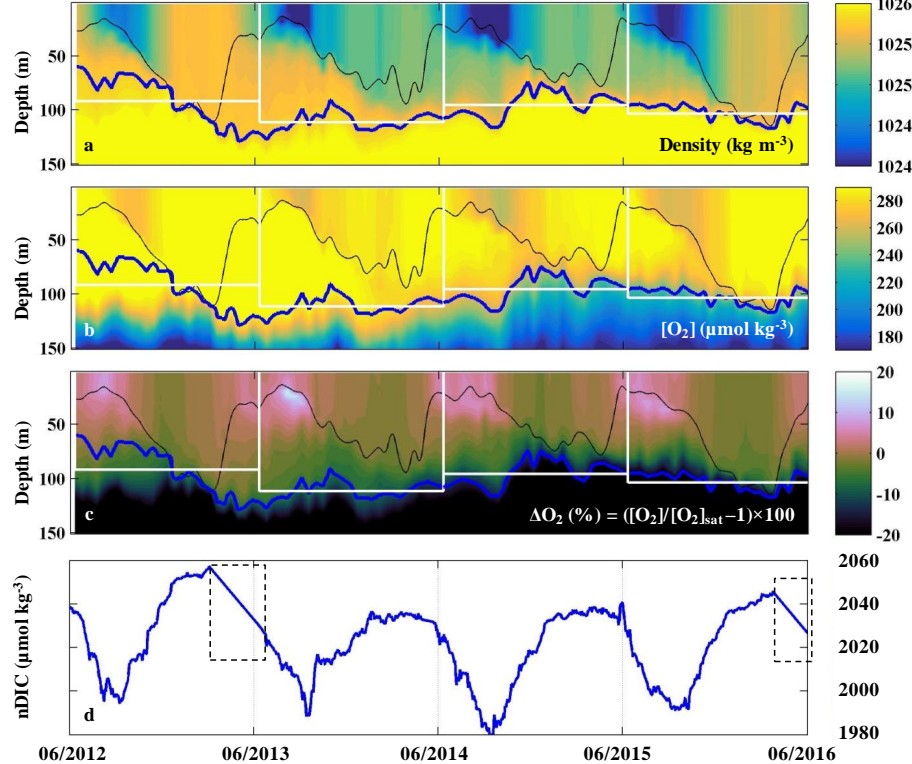

**Figure 4** (a-c) Upper ocean density, oxygen concentration, and oxygen supersaturation $\Delta O_2$ (%)
from the SOS-Argo float at OSP. The thin black line indicates the mixed layer depth, the thick
blue line indicates the pycnocline depth, and the white rectangles indicate the modeled "upper
ocean" for each of the four years that ANCP were calculated. (d) Mixed layer DIC normalized to
a surface salinity at OSP (S= 32.5) from June 2012 to June 2016. Dash line boxes indicate
periods when the pCO2 data were not available and thus were filled with a straight line
interpolation.




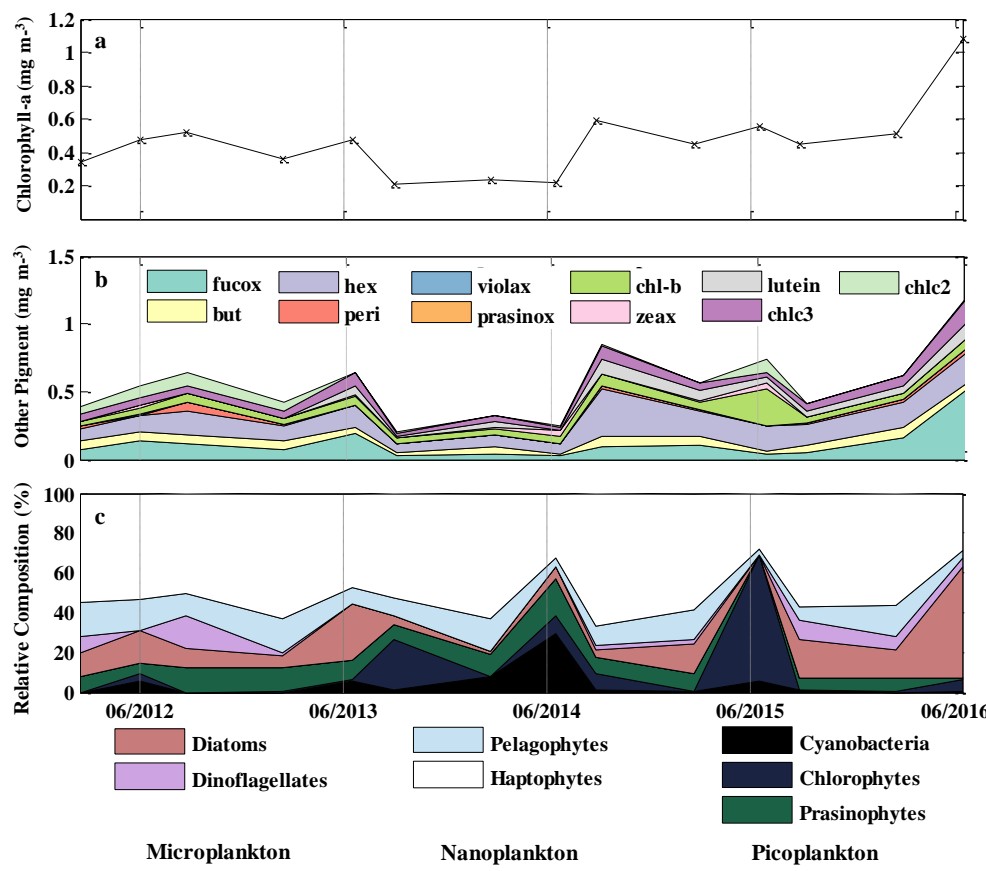

**Figure 5** Mixed layer mean (a) chl-*a* concentration (mg m$^{-3}$), (b) other pigment concentration
(mg m$^{-3}$), and (c) relative phytoplankton composition (%) at OSP. Values were determined from
HPLC pigment analysis of samples collected in February, June, and August for each year from
2012 to 2016.





**Table 1.** Pigment:Chl $a$ ratios for eight algal groups: (a) CHEMTAX initial ratio matrix, and (b) ranges of final pigment ratios obtained by CHEMTAX on the pigment data.

| | Chl $c_3$ | Chl $c_2$ | Peri | But | Fuco | Pras | Viola | Hex | Allo | Zea | Lut | Chl $b$ | Chl $a$ |
|---|---|---|---|---|---|---|---|---|---|---|---|---|---|
| (a) | | | | | | | | | | | | | |
| Cyano | 0 | 0 | 0 | 0 | 0 | 0 | 0 | 0 | 0 | 0.64 | 0 | 0 | 1 |
| Chloro | 0 | 0 | 0 | 0 | 0 | 0 | 0.049 | 0 | 0 | 0.032 | 0.17 | 0.32 | 1 |
| Prasino | 0 | 0 | 0 | 0 | 0 | 0.25 | 0.054 | 0 | 0 | 0.058 | 0.021 | 0.73 | 1 |
| Crypto | 0 | 0.2 | 0 | 0 | 0 | 0 | 0 | 0 | 0.38 | 0 | 0 | 0 | 1 |
| Diatoms | 0.08 | 0.28 | 0 | 0 | 0.99 | 0 | 0 | 0 | 0 | 0 | 0 | 0 | 1 |
| Dinofla | 0 | 0.22 | 0.56 | 0 | 0 | 0 | 0 | 0 | 0 | 0 | 0 | 0 | 1 |
| Pelago | 0.22 | 0 | 0 | 0.64 | 0.772 | 0 | 0 | 0 | 0 | 0 | 0 | 0 | 1 |
| Hapto | 0.18 | 0.21 | 0 | 0.039 | 0.289 | 0 | 0 | 0.47 | 0 | 0 | 0 | 0 | 1 |
| (b) | | | | | | | | | | | | | |
| Cyano | 0 | 0 | 0 | 0 | 0 | 0 | 0 | 0 | 0 | 0.48-0.85 | 0 | 0 | 1 |
| Chloro | 0 | 0 | 0 | 0 | 0 | 0 | 0.02-0.15 | 0 | 0 | 0.03-0.04 | 0.06-0.21 | 0.26-0.45 | 1 |
| Prasin | 0 | 0 | 0 | 0 | 0 | 0.04-0.23 | 0.02-0.06 | 0 | 0 | 0.02-0.06 | 0.017-0.022 | 0.72-1.12 | 1 |
| Crypto | 0 | 0.15-0.23 | 0 | 0 | 0 | 0 | 0 | 0 | 0.34-0.44 | 0 | 0 | 0 | 1 |
| Diatoms | 0.05-0.09 | 0.21-0.3 | 0 | 0 | 0.8-1.15 | 0 | 0 | 0 | 0 | 0 | 0 | 0 | 1 |
| Dinofla | 0 | 0.19-0.26 | 0.45-0.64 | 0 | 0 | 0 | 0 | 0 | 0 | 0 | 0 | 0 | 1 |
| Pelago | 0.11-0.25 | 0 | 0 | 0.68-1.15 | 0.22-0.82 | 0 | 0 | 0 | 0 | 0 | 0 | 0 | 1 |
| Hapto | 0.05-0.22 | 0.16-0.26 | 0 | 0.037-0.068 | 0.07-0.25 | 0 | 0 | 0.58-0.81 | 0 | 0 | 0 | 0 | 1 |

Abbreviations: Cyano, cyanobacteria; Chloro, chlorophytes; Prasino, prasinophytes; Crypto, cryptophytes; Dinofla, dinoflagellates; Pelago, pelagophytes; Hapto, haptophytes; Chl $c_3$, chlorophyll $c_3$; Chl $c_2$, chlorophyll $c_2$; Peri, peridinin; But, 19'-butanoyloxyfucoxanthin; Fuco, fucoxanthin; Pras, prasinoxanthin; Viola, violaxanthin; Hex, 19'-hexanoyloxyfucoxanthin; Allo, alloxanthin; Zea, zeaxanthin; Lut, lutein; Chl $b$, chlorophyll $b$; Chl $a$, chlorophyll.





**Table 2** Annual net community production (ANCP) determined from (a) $O_2$ mass balance, and (b) DIC mass balance. The annually integrated fluxes for each of the important terms (columns 4-9) indicate that the air sea flux and biological production terms dominate for both tracers. Two ANCP values are given in (a): one integrated from the ocean surface to the depth of annual mean pycnocline (column 3), ANCP, and another value integrated over the depth of the mixed layer, $ANCP_{mixed\ layer}$. Only the former is a measure of the biological organic carbon that escapes the upper ocean on an annual basis (see text).

**a**

| Year | Time Period (June to June) | $h$ (m) | Annual oxygen mass balance (mol $O_2$ m$^{-2}$ yr$^{-1}$) $dh[O_2]/dt = F_{A-W} + F_E + F_{Kz} + F_v + J_{NCP}$ | | | | | | $ANCP = J_{NCP}/1.45$ (mol C m$^{-2}$ yr$^{-1}$) | $ANCP_{mixed\ layer}$ (mol C m$^{-2}$ yr$^{-1}$) |
|---|---|---|---|---|---|---|---|---|---|---|
| | | | $dh[O_2]/dt$ | $F_{a-w} = F_s + F_b$ | $F_E$ | $F_{Kz}$ | $F_v$ | $J_{NCP}$ | | |
| 1 | 2012-13 | 91 | -0.7 | -2.9 | 0 | -0.6 | -0.6 | 3.5 | 2.4 ± 0.6 | 3.4 |
| 2 | 2013-14 | 111 | -1.3 | -1.5 | 0 | -0.8 | -0.2 | 1.2 | 0.8 ± 0.4 | 1.3 |
| 3 | 2014-15 | 95 | -0.6 | -1.7 | 0 | -0.9 | -1.0 | 3.0 | 2.1 ± 0.4 | 2.3 |
| 4 | 2015-16 | 103 | 0.8 | -0.1 | 0 | -0.7 | -0.3 | 2.3 | 1.6 ± 0.4 | 2.3 |

**b**

| Year | Time Period (June to June) | $h$ (m) | Annual DIC mass balance (mol C m$^{-2}$ yr$^{-1}$) $dh[DIC]/dt = F_{A-W} + F_E + F_{Kz} + F_v + J_{NCP}$ | | | | | | $ANCP = -J_{NCP}$ (mol C m$^{-2}$ yr$^{-1}$) |
|---|---|---|---|---|---|---|---|---|---|
| | | | $dh[DIC]/dt$ | $F_{a-w}$ | $F_E$ | $F_{Kz}$ | $F_v$ | $J_{NCP}$ | |
| 1 | 2012-13 | 91 | -0.2 | 1.0 | 0 | 0.7 | 0.1 | -2.0 | 2.0 |
| 2 | 2013-14 | 111 | -0.1 | 1.5 | 0 | 0.4 | 0.1 | -2.1 | 2.1 |
| 3 | 2014-15 | 95 | 0.05 | 2.0 | 0 | 0.5 | 0.1 | -2.6 | 2.6 |
| 4 | 2015-16 | 103 | -0.04 | 2.0 | 0 | 0.9 | 0.1 | -3.0 | 3.0 |





**Table 3** ANCP calculated from $O_2$ mass balance with different start dates to determine if the chosen annual period affects the conclusions (see text).

| Start Time | | 6/10/12 | 7/10/12 | 8/10/12 |
|---|---|---|---|---|
| | 1st year (2012-13) | 2.4 | 2.3 | 2.4 |
| ANCP | 2nd year (2013-14) | 0.8 | 0.9 | 0.7 |
| (mol C m$^{-2}$ yr$^{-1}$) | 3rd year (2014-15) | 2.1 | 2.6 | 2.5 |
| | 4th year (2015-16) | 1.6 | - | - |




**Table 4** Comparisons of ANCP measured with $O_2$ mass balance and ANCP predicted from the temperature dependence parameterization of planktonic metabolism using parameters from the Arctic Ocean [*Regaudie-De-Gioux and Duarte*, 2012]. Gross primary production (GPP) is calculated from ANCP in year 1 and Equation 7, and it is assumed to be the same through years 1 – 4. $ANCP_{diff} = 2.4$ (mol C m-2 $yr^{-1}$) – $ANCP_{Predicted\ or\ Measured}$

| Year | Mean temperature (°C) | **ANCP** (mol C $m^{-2}$ $yr^{-1}$) | | **ANCP$_{diff}$** (mol C $m^{-2}$ $yr^{-1}$) | |
|------|------|------|------|------|------|
| | | Predicted | Measured | Predicted | Measured |
| 1 | 8.4 | - | 2.4 | - | - |
| 2 | 10.4 | 1.9 | 0.8 | -0.5 | -1.6 |
| 3 | 10.8 | 1.9 | 2.1 | -0.5 | -0.3 |
| 4 | 9.9 | 2.1 | 1.6 | -0.3 | -0.8 |