# Peer review of "The Effect of the 2013-2016 High Temperature Anomaly in the Subarctic Northeast Pacific"

_Biogeosciences, 2018_

## Referee Comment (RC1) · Anonymous Referee #1 · 2 Aug 2018

I read the paper "The effect of the 2013-2016 high temperature anomaly in the subarctic northeast Pacific (the "Blob") on net community production" by Yang et al. It is a good paper, and I have enjoyed reading it. The paper presents the estimates of annual net community production based on the mass balance of oxygen and dissolved inorganic carbon from floats. The authors found that their annual net community production estimates are consistent with those in previous studies. The annual net community production derived from oxygen mass balance shows a decreasing trend during the "Blob". By ruling out the potential influence of horizontal advection and temperature on the annual net community production, the authors hypothesized that the decreasing trend in annual net community production likely originates from the change in phytoplankton

community structure based on HPLC measurements. Specifically, during the "Blob" the ecosystem is dominated by small phytoplankton, as a result displaying low export efficiency. This is definitely an important paper, considering: (1) the importance of net community production or export production in global carbon cycle; and (2) net community production can be derived based on the mass balance of oxygen concentration measurements from floats, which are exponentially increasing in the world's ocean. The derivation of net community production from floats is a challenging task, because of not well constrained processes controlling the variability of oxygen in the surface ocean (e.g., air-sea gas exchange).

Major comments: 1. The authors derived annual net community production using the mass balance of oxygen and dissolved inorganic carbon. The estimate based on dissolved inorganic carbon is higher than that based on oxygen concentration. The authors argue that this difference stems from residence time scale and horizontal advection. Since the model does not take into account horizontal advection, the latter makes sense to me. However, I cannot catch the explanation of residence time scale. I am not sure whether I am right. For example, if there is no gas exchange (very long equilibrium time scale), dissolved inorganic carbon balance still reflects net community production according to equation (2)?

2. According to the estimates based on oxygen mass balance, the annual net community production decreases during the "Blob". According to the authors' analysis, this decrease is likely attributable to the shift in phytoplankton community structure. I am thinking, whether can we dig a little deeper. Fundamentally, the net community production is controlled by light and nutrient availability on bottom-up control. So, I am wondering, whether we could find a connection to photosynthetically active radiation, mixed layer depth, and nutrients (e.g., nitrate). Probably, the authors have already analyzed those.

Minor comments: Lines 23-27: please see major comment 1. Line 28: shows? Line 121: I am curious whether it depends on oxygen concentration gradient. I am thinking

about this, because net community production is derived from oxygen concentration mass balance. Line 157: Do we need more runs of Monte Carlo simulation? Line 171: The assumption of no net dissolved inorganic carbon change is also related to no change in interannual variability in winter mixing? Or, there is no change in winter mixed layer depth?

---

## Referee Comment (RC2) · Anonymous Referee #2 · 17 Aug 2018

This paper gathered results of comprehensive measurements on biogeochemical and biological changes in OSP associated to the appearance of the "Blob." The authors skillfully estimated temporal variation of net community production by compiling the data that comes from various state-of-the-art platforms and sensors deployed among OSP. I agree most of their model calculation and their obtained ANCP estimation. Observed ANCP discrepancies dependent on the materials used (ca. oxygen and/or DIC) are acceptable, when considering their assumption that ignores DIC variation between mixed layer depth and pycnocline. However, I feel that there are several insufficiencies in the present manuscript on their interpretation of the results.

[Figure]

[major comment] 1. They attributed the observed ANCP decrease between 2012-13 and 2013-14 to the changes in gross primary production, by the process of elimination. However, there is no direct evidence in the present manuscript that the estimated decrease of gross primary production is caused by the low plankton biomass observed in 2013-14. Plankton biomass may regulate production rate, but this does not necessarily mean reduction of ANCP. In principle, ANCP is regulated by annual availability of limitation factor, likely iron in the case of OSP. If the amount of available iron had been same in each year, low plankton biomass in 2013-2014 would have diminished rate of primary production, but that simultaneously enhanced duration of high-production period by postponing the timing of iron exhaustion, and integrated amount of gross primary production would become just same amount. This is always true, as long as both iron availability and biological stoichiometry (Fe: O : C) are same. If the authors want to link observed low plankton biomass in 2013-14 and estimated low ANCP at that time, therefore, they need to prove that 1) there is no significant difference in iron availability during the observation period, and 2) there may be significant plankton-dependency in Fe:O:C stoichiometry.

2. The model used in this paper is only effective when there is no change of water mass during the calculation period. However, significant increase of water temperature at the emergence of the Blob implies the readers that there may be some intrusion of warm water. As not all readers are familiar with the physics of the Blob, the authors should mention briefly about that and declare continuity of water mass during the study period.

3. The authors raised several possible cause about the discrepancy between oxygen-based ANCP and that based on DIC, but none was mentioned about the inter-annual variation of POC/PIC production ratio that can make such discrepancy. Although we can understand from Figure 5 that no significant bloom of coccolithophore during the observation period, the authors should clearly mention about this in the text.

---

## Author Comment (AC1) · 23 Sep 2018

We would like to thank the reviewer for the comments. The point-to-point responses (in blue) to the reviewers' comments are listed below, and the revised texts are displayed in red.

**Reviewer 1**

Major comments:

1. The authors derived annual net community production using the mass balance of oxygen and dissolved inorganic carbon. The estimate based on dissolved inorganic carbon is higher than that based on oxygen concentration. The authors argue that this difference stems from residence time scale and horizontal advection. Since the model does not take into account horizontal advection, the latter makes sense to me. However, I cannot catch the explanation of residence time scale. I am not sure whether I am right. For example, if there is no gas exchange (very long equilibrium time scale), dissolved inorganic carbon balance still reflects net community production according to equation (2)?

In our case we are using both tracers to study the short-term (sub-annual) variations of ANCP, so gas exchange is a very important term for both tracers. As shown in Table 2b, for the DIC mass balance, gas exchange has a magnitude of about 50~80% of ANCP, which cannot be taken away from the mass balance calculation. For DIC since it has a relatively long residence time with respect to gas exchange (more than 1 year), it has already been "averaged" over a long period. On the other hand, due to the long residence time, horizontal advection also becomes more important for DIC, which causes the signal to be "averaged" over a larger spatial scale. Overall, the DIC tracer integrates temporally over many years and spatially over thousands of kilometers, and thus is a valuable tracer to get a big picture of biological carbon production over a long period or a large area without having to make many measurements. However, it may not be suitable for studying the sub-annual changes of NCP or inter-annual changes of ANCP.

2. According to the estimates based on oxygen mass balance, the annual net community production decreases during the "Blob". According to the authors' analysis, this decrease is likely attributable to the shift in phytoplankton community structure. I am thinking, whether can we dig a little deeper. Fundamentally, the net community production is controlled by light and nutrient availability on bottom-up control. So, I am wondering, whether we could find a connection to photosynthetically active radiation, mixed layer depth, and nutrients (e.g., nitrate). Probably, the authors have already analyzed those.

(1) We did analyze the influences from light and nutrient based on the available data, but they all seem to have little impacts. We added this part as Text S2 in the supplementary material.

**Text S2 Analysis of other environmental parameters that may cause ANCP variation**

(1) Shortwave radiation

Unfortunately there was no PAR sensor on the mooring (https://www.pmel.noaa.gov/ocs/sensors). On the other hand, it does have sensors measuring shortwave radiation, and the result (Figure 1, from

https://www.pmel.noaa.gov/ocs/data/fluxdisdel/) doesn't show significant changes in shortwave radiation during those four years.

[Figure]

**Figure S1** Net shortwave radiation data from Ocean Station Papa surface mooring (June 2012 to June 2016)

(2) Nutrient (nitrate) availability

Figure S2 shows the mixed layer nitrate concentration measured by an Argo near station Papa (Monterey Bay Aquarium Research institute (MBARI) Argo Float F7601, WMO # 5903714, http://www.mbari.org/science/upper-ocean-systems/chemical-sensor-group/floatviz/). There was no nutrient limit when the ANCP was the lowest in 2013-14. The nitrate concentration was near zero for a short period in the fall of 2015, but it went back in a short time and it didn't seem that nitrate is the limiting factor for biological production in this case.

[Figure]

**Figure S2** Nitrate data from MBARI Argo Float near Ocean Station Papa

(2) We define ANCP as the flux of organic carbon that escapes the "upper ocean" after a complete seasonal cycle. To be consistent with this definition NCP is integrated vertically from the surface ocean to the winter mixed layer depth (h). As shown in Table 1a, this value did vary from 91 to 111 m. To determine if this variation influences the ANCP trend, we did the calculation with a 4-year mean h value of 100 m (Line 322), and the result shows the same trend of ANCP.

(3) It is possible that the availability of iron (as mentioned by reviewer 2) played an important role in this case. See the response to reviewer 2.

Minor comments:

Lines 23-27: please see major comment 1.

See the response to major comment 1.

Line 28: shows?

Revised as suggested

Line 121: I am curious whether it depends on oxygen concentration gradient. I am thinking about this, because net community production is derived from oxygen concentration mass balance.

Revised as suggested to avoid confusion

Line 157: Do we need more runs of Monte Carlo simulation?

Our test showed that 100, 200, and 2000 runs gave similar result.

Line 171: The assumption of no net dissolved inorganic carbon change is also related to no change in interannual variability in winter mixing? Or, there is no change in winter mixed layer depth?

The change of DIC in the upper ocean includes the DIC change in the mixed layer and the DIC change between the mixed layer and the base of the defined "upper ocean" (winter mixed layer in our case), as shown in Figure R1 and Equation 1. Because we only have DIC measurements from surface mooring (mixed layer), we assume that the second term equals to zero and the annual DIC changes in the upper ocean are all from the DIC changes in the mixed layer.

$$\frac{d h[DIC]}{dt} = \frac{d\,mld[DIC]_{mld}}{dt} + \frac{d(h-mld)[DIC]_{h-mld}}{dt} \tag{1}$$

[Figure]

An annual cycle

Figure R1 A schematic of the modeled "upper ocean"

As shown in Table 1a and 1b, there were inter-annual changes in winter mixed layer depth (h), which chances from 91 to 111 m during this 4-year period. For vertical flux (Fkz and Fv) calculation, we need the DIC gradient at depth of h, which was computed from the oxygen gradients calculated from measured oxygen gradients assuming dO2/dz to dDIC/dz ratio of 1.45 (line 168).

---

## Author Comment (AC2) · 23 Sep 2018

We would like to thank the reviewer for the comments. The point-to-point responses (in blue) to the reviewers' comments are listed below, and the revised texts are displayed in red.

**Reviewer 2**

1. They attributed the observed ANCP decrease between 2012-13 and 2013-14 to the changes in gross primary production, by the process of elimination. However, there is no direct evidence in the present manuscript that the estimated decrease of gross primary production is caused by the low plankton biomass observed in 2013-14. Plankton biomass may regulate production rate, but this does not necessarily mean reduction of ANCP. In principle, ANCP is regulated by annual availability of limitation factor, likely iron in the case of OSP. If the amount of available iron had been same in each year, low plankton biomass in 2013-2014 would have diminished rate of primary production, but that simultaneously enhanced duration of high-production period by postponing the timing of iron exhaustion, and integrated amount of gross primary production would become just same amount. This is always true, as long as both iron availability and biological stoichiometry (Fe: O : C) are same. If the authors want to link observed low plankton biomass in 2013-14 and estimated low ANCP at that time, therefore, they need to prove that 1) there is no significant difference in iron availability during the observation period, and 2) there may be significant plankton-dependency in Fe:O:C stoichiometry.

Thanks for the comment. In this manuscript, we analyzed the connections between ANCP decrease and the high temperature anomaly (the "blob") in the context of multiple physical and biological processes. Our analysis showed that the sharp decreasing trend of ANCP in 2013-14 has strong connections with the phytoplankton community composition change and low plankton biomass during the same period. The changes in phytoplankton community composition and biomass could be ultimately in response to the lack of micronutrients like iron due to the enhanced stratification during the "blob", but unfortunately we do not have iron data available to confirm that.

We have revised the text in the discussion as follows:

"These changes in phytoplankton community composition could be ultimately in response to the lack of micronutrients like iron (due to enhanced stratification from the "blob" that restricted the vertical supply), which has been shown to regulate phytoplankton biomass and composition in this high-nutrient low-chlorophyll region (e.g. Hamme et al., 2010; Marchetti et al., 2006), . Unfortunately, we do not have iron data available to confirm that at this time. "

2. The model used in this paper is only effective when there is no change of water mass during the calculation period. However, significant increase of water temperature at the emergence of the Blob implies the readers that there may be some intrusion of warm water. As not all readers are familiar with

the physics of the Blob, the authors should mention briefly about that and declare continuity of water mass during the study period.

We adopted the reviewer's suggestion, and revised the text in Section 2.1.1 and supplemental materials as follows:

In section 2.1.1:

Furthermore, the temperature time series measured by the SOS-Argo (Figure S1) shows no significant intrusions of fronts/eddies, and the continuity of water mass during the study period also allows us to use this simplified model that ignores horizontal processes.

In supplemental materials:

[Figure]

**Figure S1** Upper temperature evolution measured by the SOS-Argo float at OSP, which shows the continuity of water mass during the study period (2012-2016).

3. The authors raised several possible cause about the discrepancy between oxygen based ANCP and that based on DIC, but none was mentioned about the inter-annual variation of POC/PIC production ratio that can make such discrepancy. Although we can understand from Figure 5 that no significant bloom of coccolithophore during the observation period, the authors should clearly mention about this in the text.

We adopted the reviewer's suggestion, and revised the text in Section 4.1 as follows:

Alternatively, the production ratio of particulate organic carbon (POC) and particulate inorganic carbon (PIC) may cause the inter-annual variation of DIC mass balance. However, in our case since there was no significant bloom of haptophytes (e.g. coccolithophore) during the

study period (Figure 5c), it is unlikely that the inter-annual change in POC/PIC ratio would affect

the ANCP result calculated from DIC mass balance.

---

## Author Response (AR2)

We would like to thank the reviewer for the comments. The point-to-point responses (in blue) to the reviewers' comments are listed below, and the revised texts are displayed in red.

As the authors agreed in their response that they have no direct evidence about causal relationship between 2013-2014 ANCP decrease and phytoplankton community composition change and low plankton biomass, I demand the authors to delete any mentions that suggests causal relationship between these matters.
They have clearly detected 2013-2014 ANCP decrease simultaneous change in phytoplankton community composition and plankton biomass, but this never means by themselves that the latter is the cause of the former. It is also possible that the former cause the latter, and it's even more probable that the all of these findings is a results of single, not detected event (ca, GPP decrease).

Page 2, line 34
"it was most likely due to changes in plankton community composition"
=>this should be changed to like that:
"it was likely due to some biological processes rather than physical processes."

This sentence has been changed as:

"it was most likely due to the temperature-induced changes in biological processes."

Page15, lines 314-316.
"The sharp decrease……(Figure5a)." =>delete this sentence.

This sentence has been deleted as suggested.

Page 17, lines 351-353.
"We suggest that the low phytoplankton biomass observed in the 2nd year '2013-14, Figure 5a), and the observed change in phytoplankton community composition (figure 5c) are the most likely causes for the ANCP decrease."
=>should be changed to:
"We suggest that the change in GPP are the most likely causes for the ANCP decrease."

Here you made big jump: Your elimination-method based discussion had suggested that GPP decrease must be required to explain observed ANCP decrease, but NONE in your discussion suggested that the cause of GPP decrease is reduced Chla abundance and change in plankton composition. Again, I say that these two events can be a RESULTS of GPP decrease as well as ANCP decrease, not the cause of it.

As mentioned by the reviewer, it is not appropriate to jump into the conclusion that one event is the cause of the other. So we change the first sentence of this paragraph as follows:

"Having ruled out the above likely candidates, we suggest that the observed ANCP decrease is most likely linked to the changes in GPP (e.g. low phytoplankton biomass observed in the 2nd year Figure 5a) and phytoplankton community composition (Figure 5c)."

Page 17, lines 363-368
"These changes in phytoplankton community composition could be ultimately in response to….."
=>should be changed to:

"Such change in GPP, as well as decreased Chla abundance and change in community composition could be ultimately in response to ….."

And the last part of this paragraph has been re-written as:

"These changes in GPP and phytoplankton community composition could be ultimately in response to the lack of micronutrients like iron (due to enhanced stratification from the "blob" that restricted the vertical supply), which has been shown to regulate phytoplankton biomass and composition in this high-nutrient low-chlorophyll region (e.g. Hamme et al., 2010; Marchetti et al., 2006), . Unfortunately, we do not have iron data available to confirm that at this time."

page 18, line 384-385
"the ANCP decrease was most likely due to changes in phytoplankton abundance and community composition after the "Blob" entered the area.
=>should be changed to such as:
"the ANCP decrease, as well as change in Chla abundance and phytoplankton composition, was most likely due to changes in GPP after the entry of the "Blob." The ultimate cause of such GPP decrease, however, is not be able to specified by our analysis."

The last sentence has been re-written as:

"Our analysis showed that the ANCP decrease, as well as changes in phytoplankton abundance and community composition, was most likely due to changes in GPP after the "Blob" entered the area. These changes could be ultimately in response to the lack of micronutrients like iron during the "Blob" period. However, the ultimate cause cannot be specified by our analysis at this time."